# Anti-Biofilm Effects of Synthetic Antimicrobial Peptides Against Drug-Resistant *Pseudomonas aeruginosa* and *Staphylococcus aureus* Planktonic Cells and Biofilm

**DOI:** 10.3390/molecules24244560

**Published:** 2019-12-12

**Authors:** Seong-Cheol Park, Min-Young Lee, Jin-Young Kim, Hyeonseok Kim, Myunghwan Jung, Min-Kyoung Shin, Woo-Kon Lee, Gang-Won Cheong, Jung Ro Lee, Mi-Kyeong Jang

**Affiliations:** 1Department of Polymer Science and Engineering, Sunchon National University, Suncheon, Jeonnam 57922, Korea; schpark9@gnu.ac.kr (S.-C.P.); jyfrog@hanmail.net (J.-Y.K.); hht95@naver.com (H.K.); 2Department of Microbiology, Gyeongsang National University School of Medicine, Jinju, Gyeongnam 52727, Korea; mylee365@empas.com (M.-Y.L.); mjung@gnu.ac.kr (M.J.); mkshin@gnu.ac.kr (M.-K.S.); wklee@gnu.ac.kr (W.-K.L.); 3Division of Applied Life Sciences and Research Institute of Natural Science, Gyeongsang National University, Jinju, Gyeongnam 52828, Korea; gwcheong@gnu.ac.kr; 4National Institute of Ecology, 1210 Geumgang-ro, Maseo-myeon, Seocheon-gun 33657, Korea; 5The Research Institute for Sanitation and Environment of Coastal Areas, Sunchon National University, Suncheon, Jeonnam 57922, Korea

**Keywords:** biofilm inhibition, synthetic antimicrobial peptide, drug-resistant bacteria, extracellular polymeric substances, biofilm degradation

## Abstract

Biofilm-associated infections are difficult to manage or treat as biofilms or biofilm-embedded bacteria are difficult to eradicate. Antimicrobial peptides have gained increasing attention as a possible alternative to conventional drugs to combat drug-resistant microorganisms because they inhibit the growth of planktonic bacteria by disrupting the cytoplasmic membrane. The current study investigated the effects of synthetic peptides (PS1-2, PS1-5, and PS1-6) and conventional antibiotics on the growth, biofilm formation, and biofilm reduction of drug-resistant *Pseudomonas aeruginosa* and *Staphylococcus aureus*. The effects of PS1-2, PS1-5, and PS1-6 were also tested in vivo using a mouse model. All peptides inhibited planktonic cell growth and biofilm formation in a dose-dependent manner. They also reduced preformed biofilm masses by removing the carbohydrates, extracellular DNA, and lipids that comprised extracellular polymeric substances (EPSs) but did not affect proteins. In vivo, PS1-2 showed the greatest efficacy against preformed biofilms with no cytotoxicity. Our findings indicate that the PS1-2 peptide has potential as a next-generation therapeutic drug to overcome multidrug resistance and to regulate inflammatory response in biofilm-associated infections.

## 1. Introduction

Biofilm consists of microorganisms embedded in a self-produced matrix of extracellular polymeric substances (EPSs) containing polysaccharides, extracellular DNA, proteins, and lipids [1,2,3]. The emergence of multidrug-resistant microorganisms in clinical therapeutics is a global healthcare concern. The National Institutes of Health in the USA reported that approximately 80% of chronic infections in humans are biofilm-related ones [4]. Infectious processes in biofilm are divided into two types, namely device- and non-device-associated infections [5]. Device-associated infections are caused by microbial colonization of medical devices such as urinary catheters, bone joints, heart valves, dental implants, prostheses, contact lenses, and endotracheal tubes [5,6,7,8,9,10]. The occurrence of nosocomial infections through biomaterials or implants is approximately 60–70%; of those, 720,000 cases of central line-associated bloodstream infections occur annually in the USA via dialysis and intensive care units with an associated 12% mortality and a $45,000 increase in treatment cost per episode [11]. Non-device-associated infections arise due to biofilm formation on tissues containing mucosal surfaces, epithelial tissues, and teeth [12,13,14].

Bacterial cells growing in a biofilm have physiologically different characteristics than planktonic cells of the same bacteria, and their antibiotic resistance can increase up to 1000-fold owing to the effect of self-produced EPSs. Overlong cultivation of bacterial cells imparts an adhesion ability to animal tissues and inorganic materials that enables biofilm formation. Biofilm provides a survival advantage by enhancing nutrient trapping and colonization ability compared to that of planktonic or free-floating bacteria [15]. Bacteria form biofilm in response to various factors including nutritional signals, surface attachment site recognition, exposure to sublethal antibiotic concentrations, and environmental stresses [16,17]. Generally, biofilm formation is initiated by the attachment of planktonic bacterial cells to a surface via weak van der Waals forces, followed by tighter or irreversible anchoring by pili. To facilitate the gathering and attachment of other planktonic cells, various adhesion sites are built on the matrix holding the biofilm. Bacterial cells are embedded within this EPS matrix, which is a very important target in managing biofilms and drug-resistant bacteria. During colonization, bacteria can communicate via a quorum-sensing system to facilitate biofilm development. The developed biofilms are eventually dispersed and bacteria move to other surfaces such as tissues, organs, and medical devices where these stages are repeated [16,18,19].

Pathogenic biofilm development is an important clinical concern in terms of economic losses, morbidity, and mortality; therefore, the development of anti-biofilm agents is an important factor in managing human infection. Antimicrobial peptides (AMPs) have gained increasing attention as a possible alternative to conventional drugs to combat drug-resistant microorganisms because they inhibit the growth of planktonic bacteria by disrupting the cytoplasmic membrane and inhibiting intracellular macromolecules. AMPs are considered a promising anti-biofilm agent in both drug therapies and the development of anti-infective devices. Natural AMPs are cationic, amphipathic, and 12–60 amino acids long. LL-37, a human cathelicidin peptide, has shown potent inhibitory and reductive effects against *Pseudomonas aeruginosa* biofilm [20] and inhibited both the surface attachment and development of *Staphylococcus epidermidis* biofilm [21]. P10, its analog peptide, has shown remarkable anti-biofilm activity, inhibiting biofilm development and reducing the preformed biofilm of multidrug-resistant *Staphylococcus aureus* [22]. CAMA, a hybrid peptide containing an N-terminus of cecropin A and melittin, effectively inhibited methicillin-resistant *S. aureus* (MRSA) biofilm formation [23]. Several anti-biofilm mechanisms have been discovered and include the membrane potential disruption of biofilm-embedded bacterial cells [24], the interruption of quorum-sensing systems [20], EPS reduction [25,26], alarmone system inhibition [27], and downregulation of genes related to biofilm formation and binding protein transportation [28]. Although many anti-biofilm peptides have been designed and investigated, further studies identifying and defining key amino acids and/or structural features responsible for biofilm prevention and eradication are needed.

In a previous study, we reported a potent antimicrobial activity and membranolytic mechanism of a novel peptide series with repeated sequences of “XWZX” (X: lysine or arginine, Z: leucine, tyrosine, valine, or glycine) [29]. Among eight peptides, PS1-2 ((KWYK)_3_) peptide with the highest therapeutic index (cell selectivity between bacteria and mammalian cells) was selected as an anti-biofilm target peptide in this study. To determine which of the lysine and arginine residues as a cationic amino acid is effective for anti-biofilm activity, PS1-5 ((RWYR)_3_) was compared. In addition, although PS1-5 ((KWLK)_3_) was found to be highly cytotoxic, it was chosen to compare the effects of aliphatic and aromatic side chains on anti-biofilm prevention. Three peptides synthesized by using a microwave peptide synthesizer were purified by C_18_ reversed-phase HPLC. Their anti-biofilm activity on drug-susceptible and drug-resistant *P. aeruginosa* and *S. aureus* cells was proved by *in vitro* phenotypic and EPS analyses, and by *in vivo* evaluation using a catheter-implanted mouse model.

## 2. Results and Discussion

### 2.1. Growth Inhibition of Planktonic Cells by Peptides

The planktonic cell growth of five *P. aeruginosa* and *S. aureus* strains was evaluated to determine the minimum inhibitory concentration (MIC) of PS1-2, PS1-5, PS1-6, and conventional antibiotics. As expected, the two conventional antibiotics (gentamicin and oxacillin) inhibited the propagation of drug-susceptible *P. aeruginosa* at a very low concentration; however, their MICs increased remarkably for drug-resistant *P. aeruginosa* (Table 1). All peptides showed potent antimicrobial activity, with MICs ranging from 4 to 1 µM against five *P. aeruginosa* strains. Table 1 shows that PS1-2 and PS1-6 peptides significantly inhibited the growth of all tested *S. aureus* strains. Interestingly, MIC values of the PS1-5 peptide were 32 µM for *S. aureus* ATCC 25923, 16 µM for *S. aureus* CCARM 3125 and 3709, and 2 µM for *S. aureus* DRSa 3399 and 3518, indicating that PS1-5 was more active against drug-resistant *S. aureus* cells than drug-susceptible cells. The arginine residue of PS1-5 binds strongly with anionic bacterial surfaces and fatty acids of phospholipid as it possesses an aliphatic straight chain that ends in a guanidino group that is protonated to result in the guanidinium form at physiological pH. Moreover, it was found to promote the translocation of cell-penetrating peptides by guanidinium groups [30,31,32,33]. Therefore, this result may be due to an increase in peptide-binding affinity through cell-surface alterations of drug-resistant strains.

### 2.2. Inhibitory Kinetics of Peptides on Bacterial Growth

In order to investigate the sustained AMP action, *P. aeruginosa* and *S. aureus* growth rates were investigated for 24 h in the presence of PS1-2, PS1-5, PS1-6, gentamicin, erythromycin, or oxacillin at their MICs (Figure 1). During 24 h of peptide incubation, the growth of drug-susceptible *P. aeruginosa* (ATCC 15692) and *S. aureus* (ATCC 25923) was significantly and continuously inhibited by all tested AMPs and antibiotics (Figure 1a,c, respectively). Figure 1b shows that gentamicin and oxacillin did not inhibit *P. aeruginosa* CCARM 2073 growth, and the growth of *S. aureus* CCARM 3125 cells was not inhibited at 128 µM of erythromycin and 256 µM of oxacillin (Figure 1d); however, the growth of all tested bacterial strains was not detected for 24 h in the presence of PS1-2, PS1-5, or PS1-6. These results indicate that AMPs can prevent the proliferation of both drug-resistant and drug-susceptible bacteria owing to their quick and efficient bactericidal mechanisms.

### 2.3. Biofilm Formation Inhibition

The effects of peptides in biofilm formation for *P. aeruginosa* (ATCC 15692, CCARM 2073, and DRPa 4007) and *S. aureus* (ATCC 25923, CCARM 3125, and DRSa 3518) using the crystal violet biomass staining method are shown in Figure 2. Oxacillin and erythromycin significantly inhibited the biofilm formation of drug-susceptible *P. aeruginosa* (ATCC 15692) and *S. aureus* (ATCC 25923) cells in a dose-dependent manner, respectively, at from 2 to 16 µM.; however, biofilm formation by drug-resistant strains (CCARM 2073 and DRPa 4007) was inhibited less than 20% at a 64 µM concentration. In contrast, all tested strains were inhibited more than 50% at an 8 µM concentration in the presence of all peptides. We suggest that these biofilm-inhibitory effects may be due to inhibiting bacterial growth via bacteriocidal action of peptides or preventing attachment of bacterial cells on plate surface via cationic property of peptides.

### 2.4. Reductive Effects of Peptides on Preformed Biofilms

After 24 h of biofilm formation, different antimicrobial samples (PS1-2, PS1-5, PS1-6, gentamycin, oxacillin, or erythromycin) were added to evaluate the reductive activity on preformed biofilms, followed by an additional 24 h of incubation. In this assay, the amount of the treated bacteria was increased 10-fold, compared to the above inhibition assay for biofilm formation, in order to completely form a biofilm. In addition, it was required to remove the biofilm EPSs and bacteria on the well surface. Therefore, a large quantity of peptides was exposed in the preformed biofilm. As shown in Figure 3, three peptides significantly contributed to reducing the preformed biofilm in both *P. aeruginosa* and *S. aureus* strains. PS1-2 had the best removal activity, showing 43.87% and 65.6% biofilm reduction in *P. aeruginosa* CCARM 2073 and DRPa 4007, and 60.6% and 59.54% biofilm reduction in *S. aureus* CCARM 3125 and DRSa 3518, respectively, at a concentration of 16 µM.

### 2.5. Effect of Peptides on Biofilm Components

To investigate how peptide reduces bacterial biofilms, specific fluorescent dyes that are able to bind to fluorescein isothiocyanate-labeled concanavalin A (FITC-ConA) for carbohydrates—Nile red for lipids, DAPI for extracellular DNA, and SYPRO red for proteins—were applied after 24 h of incubation with peptides, then visualized using fluorescence microscopy (Figure 4a) and quantified using fluorescence spectrophotometry (Figure 4b). Both sets of data indicated that peptides reduced the biofilm biomass by breaking down the network structures of carbohydrates, lipids, and extracellular DNA via their amphipathic structure and cationicity.

### 2.6. Biofilm Reduction in the Presence of Peptides

After treatment of preformed *P. aeruginosa* and *S. aureus* biofilms on a plastic disk, biofilm biomass was observed using SEM. Biofilms and bacteria were significantly reduced in the presence of PS peptides (Figure 5). The SEM images showed a significant volume of *P. aeruginosa* DRPa 4007 biofilm (Figure 5a, control) and *S. aureus* DRSa 3125 biofilm (Figure 5b, control) in the absence of peptides compared to those in the presence of peptides. Oxacillin- or erythromycin-treated disks showed massive biofilm formation because drug-resistant bacterial strains were used for biofilm formation.

### 2.7. In Vivo Anti-Biofilm Action of PS Peptides

To investigate the in vivo anti-biofilm activity of PS peptides, biofilm that was preformed for 36 h on catheter samples was subcutaneously implanted into the back of nude mice (Figure 6a), followed by injection of peptides twice daily for two days. In the untreated group, three of the four mice died within two days. One mouse survived for five days, but displayed severely swollen and blackened back skin (data not shown). Increased epidermal thickness, dermis destruction, and immune cell numbers (blue arrow in Figure 6b) were observed in the presence of peptides by histological analysis. Interestingly, all mice treated with PS peptides survived. The skin tissue of mice treated with PS1-2 peptide (Figure 6b, iii) recovered similarly to epidermal and dermis thickness of the control mice (Figure 6b, i). Although mice treated with PS1-5 and PS1-6 peptides survived and their skin did not appear seriously injured, the increased epidermal thickness and cell necrosis (red arrow in Figure 6b) were observed by H&E staining (Figure 6b, iv and v, respectively). This is due to the previously reported cytotoxic effects of PS1-5 and PS1-6 peptides in normal cells [29]. Based on these findings, PS1-2 (KWYK)_3_ may be an interesting candidate for future drug development strategies in the field of biofilm-associated infections, as it displayed no cytotoxicity in in vitro hemolytic and cytotoxic assays.

## 3. Materials and Methods

### 3.1. Materials

Oxacillin, erythromycin, gentamycin, glutaraldehyde, fluorescein isothiocyanate-labeled concanavalin A (FITC-ConA), SYPRO red, Nile red, and 4′,6-diamidino-2-phenylindole (DAPI) were obtained from Sigma-Aldrich Co. (St. Louis, MO, USA). 9-Fluorenylmethoxycarbonyl (Fmoc) amino acids and Oxyma pure were purchased from CEM Co. (Matthews, NC, USA). Diisopropylcarbodiimide (DIC) and crystal violet were obtained from Tokyo Chemical Industry Co., Ltd. (Tokyo, Japan). All other chemicals and solvents were of analytical or reagent grade and used as received.

### 3.2. Peptide Synthesis by Solid-Phase Method

Microwave-assisted automated solid-phase peptide synthesis (Liberty Blue CEM Corporation, Matthews, NC, USA) was used to synthesize PS1-2, PS1-5, and PS1-6 (PS1-2: KWYKKWYKKWYK-CONH_2_, PS1-5: RWYRRWYRRWYR-CONH_2_, and PS1-6: KWLKKWLKKWLK-CONH_2_). Rink Amide resin (Novabiochem) (0.55 mmol/g) was used and Fmoc deprotection was assessed by 20% piperidine in dimethylformamide (DMF). Each coupling step of Fmoc amino acids was achieved using microwave heating in the presence of DIC and Oxyma pure in DMF. Resin-synthesized peptide was transferred to a conical tube, washed with dichloromethane, and allowed to air-dry. Peptide cleavage was performed by treatment with trifluoroacetic acid (TFA)/triisopropylsilane/DiH_2_O (95:2.5:2.5, *v*/*v*/*v*) for 2 h at room temperature. The cleaved peptide–TFA solution was precipitated with diethyl ether and then dried under a vacuum pump (Edwards RV5, Seongnam-si, Korea) to obtain a powder. The synthesized peptides were purified using a Zorbax C_18_ column (21.2 × 250 mm, 300 Å, 7 μm) on a Shimadzu Preparative HPLC system (Kyoto, Japan) using 5%–60% acetonitrile gradient in water with 0.05% TFA. Molecular masses were confirmed using a matrix-assisted laser desorption ionization mass spectrometer (MALDI II, Kratos Analytical Ltd., Manchester, UK) [34].

### 3.3. Bacterial Strains and Growth Conditions

The anti-biofilm assay was performed using *P. aeruginosa* PAO_1_ (ATCC 15692) and *S. aureus* (ATCC 25923) obtained from the American Type Culture Collection (ATCC, Manassas, VA, USA). Drug-resistant *P. aeruginosa* (CCARM 2073 and CCARM 2075) and *S. aureus* (CCARM 3125 and 3709) were purchased from the Culture Collection of Antibiotic Resistant Microbes (CCARM, Seoul, Korea). Drug-resistant *P. aeruginosa* ((DRPa)-4007 and DRPa-3241) and drug-resistant *S. aureus* ((DRSa)-3399 and DRSa-3518) were clinically isolated from patients with otitis media. All strains were grown in Mueller–Hinton (MH) broth under aerobic conditions at 37 °C.

### 3.4. Biofilm Susceptibility Assay

#### 3.4.1. Growth Inhibition in Planktonic Bacterial Cells

Bacterial cells were cultured at 37 °C in MH broth and the antimicrobial activities of peptides for planktonic bacterial cells were determined using micro-dilution assays. Briefly, bacteria collected in the mid-log growth phase were suspended in 10 mM sodium phosphate (pH 7.2) containing 20% (*v*/*v*) brain heart infusion (BHI) broth. Two-fold serial dilutions of each peptide with concentrations ranging from 0.5 to 64 μM were added to sterile 96-well plates, after which 50 µL of the cell suspension (1 × 10^6^ colony forming units (CFU)/mL) were added to each well. After incubation at 37 °C for 24 h, the turbidity of each well was measured by absorbance at 600 nm using a SpectraMax M5 Microplate Reader (Molecular Devices, Sunnyvale, CA, USA). The lowest concentration of peptide that completely inhibited bacterial growth was defined as the MIC; MIC values were calculated as an average of several independent experiments conducted in triplicate [35].

#### 3.4.2. Inhibition of Biofilm Formation Assay

The inhibitory effect of peptides in bacterial biofilm formation was determined using 96-well plates. Pregrown bacteria were added to peptide solutions at concentrations ranging from 0.5 to 64 µM in BHI broth supplemented with 2% (*w*/*v*) sucrose at a final concentration of 1 × 10^6^ CFU/mL. After incubation at 37 °C for 24 h, the medium was discarded and the wells were washed with phosphate-buffered saline (PBS). The biofilm was fixed with methanol for 15 min and air-dried at room temperature, then stained with 0.1% crystal violet for 5 min. The wells were rinsed with water, then 200 µL of 95% ethanol was added to each well and the plate was shaken at room temperature for 30 min. Absorbance was measured at 570 nm on a SpectraMax M5 Microplate Reader (Molecular Devices, Sunnyvale, CA, USA) to determine biofilm biomass. Each assay was performed in triplicate.

#### 3.4.3. Reductive Assay in Preformed Biofilm

Bacteria suspended in BHI broth supplemented with 2% (*w*/*v*) sucrose were adjusted to a density of 1 × 10^7^ CFU/mL and seeded in a 96-well flat-bottomed plate, followed by incubation for 48 h at 37 °C. The wells were washed carefully with PBS, and peptides ranging from 250 to 1.95 µM were added. After incubation for 24 h at 37 °C, the methods used for the inhibitory assay described in Section 3.4.2 were followed. Each assay was performed in triplicate.

### 3.5. Growth Inhibition Kinetics

Suspensions of *P. aeruginosa* PAO_1_ ATCC 15692 and CCARM 2073 (1 × 10^6^ CFU/mL) were added to peptide solutions in 1.5 mL microtubes. Bacterial cells were exposed to each of the peptides at their MICs for 0, 0.5, 1, 2, 3, 4, 6, 8, 12, 14, 16, 20, 22, and 24 h. The samples were diluted 20-fold, then plated on MH agar and incubated overnight at 37 °C before counting the colonies [35].

### 3.6. Reductive EPS Analyses

Fluorescence microplate and microscopic analyses were performed to visualize the removed EPS components. After following the processes described in Section 3.4.3, the peptide-treated biofilm was washed with PBS and stained using four fluorescent dyes (100 µL of 0.5 µg/mL DAPI for nucleic acids, 100 µL of 1/500 diluent of SYPRO red for proteins, 100 µL of 20 µM Nile red for lipids, and 100 µL of 50 µg/mL FITC-ConA for carbohydrates) [36,37,38].

### 3.7. Scanning Electron Microscopy

Biofilm formation occurred on plastic disks (12 mm dimension, 188 µm thick, SPL life science, Pocheon-si, Korea) for 36 h, and peptides at each MIC were incubated for 24 h. The plastic films were washed in PBS and prefixed in 1 mL of 5% (*v*/*v*) glutaraldehyde in PBS for 2 h. The samples were then washed again in PBS and subsequently post-fixed in 1% osmium tetroxide (Electron Microscopy Sciences, Washington, PA, USA) for 1 h. The samples were washed again in PBS, then dehydrated using the OTTIX shaper (Diapath S.p.A, Bergamo, Italy), dried using hexamethyldisilazane, and sputter-coated with 10 nm of gold. All samples were observed and photographed on an FESEM (JSM-7100F, JEOL Ltd., Tokyo, Japan) [39].

### 3.8. In Vivo Study

The in vivo animal study was performed in accordance with the protocols and guidelines approved by the Institutional Animal Care and Use Committee (IACUC) of Sunchon National University (SCNU IACUC-2019-10). Female athymic NCr-nu/nu nude mice (7 weeks old) were obtained from Koatech Co. (Pyongtaek, Gyeonggido, South Korea). Mice (four per cage and four per group) were kept in ventilated cages at 22–25 °C with 50–60% humidity and a 12/12 h light/dark cycle. They were fed standard food (Teklad Laboratory diet for rodents) and distilled water. *S. aureus* DRSa 3125 was suspended in BHI broth supplemented with 2% (*w*/*v*) sucrose (1 × 10^7^ CFU/mL) and seeded in a 24-well flat-bottomed plate. Catheter samples (IV catheter, 16 Gauge, BD KOREA, Seoul, Korea) cut to 1 cm were added, followed by incubation for 36 h at 37 °C. Catheter samples with preformed biofilm were implanted into the skin on the back of the test mice. Mice were anesthetized by inhalation of 5% (induction) and 2% (maintenance) isoflurane in pure oxygen. DERMABOND (Ethicon, Somerville, NJ, USA) was used for wound closure. After 12 h, peptides were injected into the lumen of the catheter with 1 mg/kg (one injection each day for two days). Five days after the last peptide injection, untreated and treated animal groups were euthanized with CO_2_ and the tissues around the catheter were harvested. Histological analysis was performed by hematoxylin and eosin (H&E) staining. Briefly, tissues were washed with PBS, followed by fixation in 4% paraformaldehyde for 24 h at 4 °C, dehydration through a 50–100% ethanol series for 2 h each, and bathing three times in xylene for 20 min each. The paraffin-embedded samples were sectioned at a thickness of 5 µm (Leica microtome, Deerfield, IL, USA), followed by H&E staining and microscopic observation (IX71, Olympus, Tokyo, Japan).

### 3.9. Statistical Analysis

The mean values of at least three independent determinations ± SD and the *p*-values (Student’s *t*-test) were calculated using Excel software.

## 4. Conclusions

In summary, this study showed that the de novo designed AMPs PS1-2, PS1-5, and PS1-6 have an inhibiting effect on biofilm formation and disrupting activity for preformed biofilms in drug-susceptible and drug-resistant *P. aeruginosa* and *S. aureus* strains. Moreover, all PS peptides reduced the biofilm by detaching the carbohydrates, nucleic acids, and lipids of the EPS but did not detach the proteins. Future studies should focus on the use of PS1-2 as treatment for antibiotic resistant bacteria because it shows good efficacy without the risk of cytotoxicity.

## Figures and Tables

**Figure 1 molecules-24-04560-f001:**
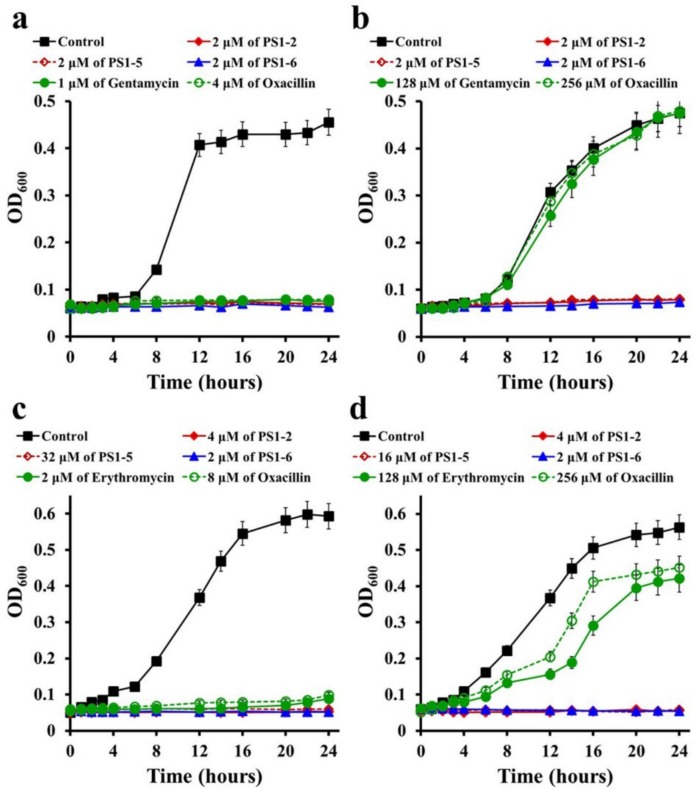
Growth-inhibitory kinetics of peptides and antibiotics against *Pseudomonas aeruginosa* (**a**) ATCC 15692 and (**b**) CCARM 2073, and *Staphylococcus aureus* (**c**) ATCC 25923 and (**d**) CCARM 3125 strains.

**Figure 2 molecules-24-04560-f002:**
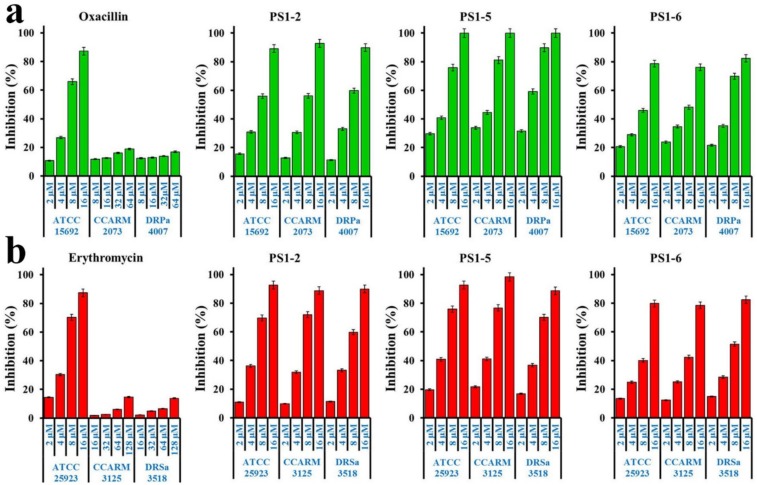
Inhibitory activity of peptides and antibiotics on biofilm formation of (**a**) *P. aeruginosa* and (**b**) *S. aureus* bacterial cells. (**a**) Oxacillin was treated at from 2 to 16 µM for *P. aeruginosa* ATCC 15692 and from 8 to 64 µM for *P. aeruginosa* CCARM 2073 and DRPa 4007. (**b**) Erythromycin was treated at from 2 to 16 µM for *S. aureus* ATCC 25923 and from 16 to 128 µM for *S. aureus* CCARM 3125 and DRSa 3518. Peptides ranged from 2 to 16 µM and were incubated in all tested bacteria.

**Figure 3 molecules-24-04560-f003:**
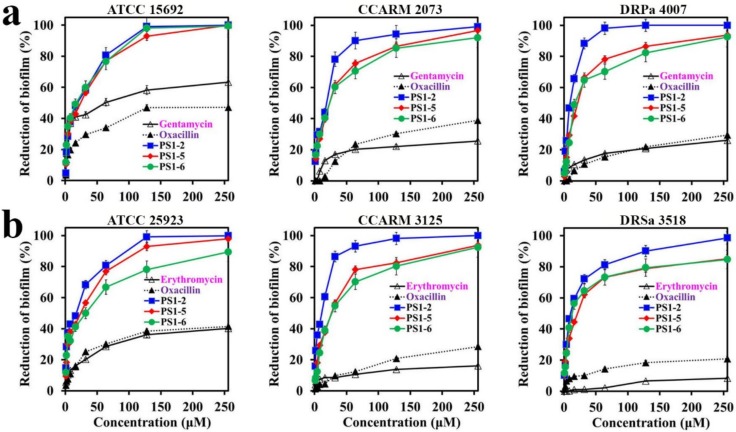
Reductive effect of peptides and antibiotics against preformed biofilm in (**a**) *P. aeruginosa* and (**b**) *S. aureus* bacterial cells.

**Figure 4 molecules-24-04560-f004:**
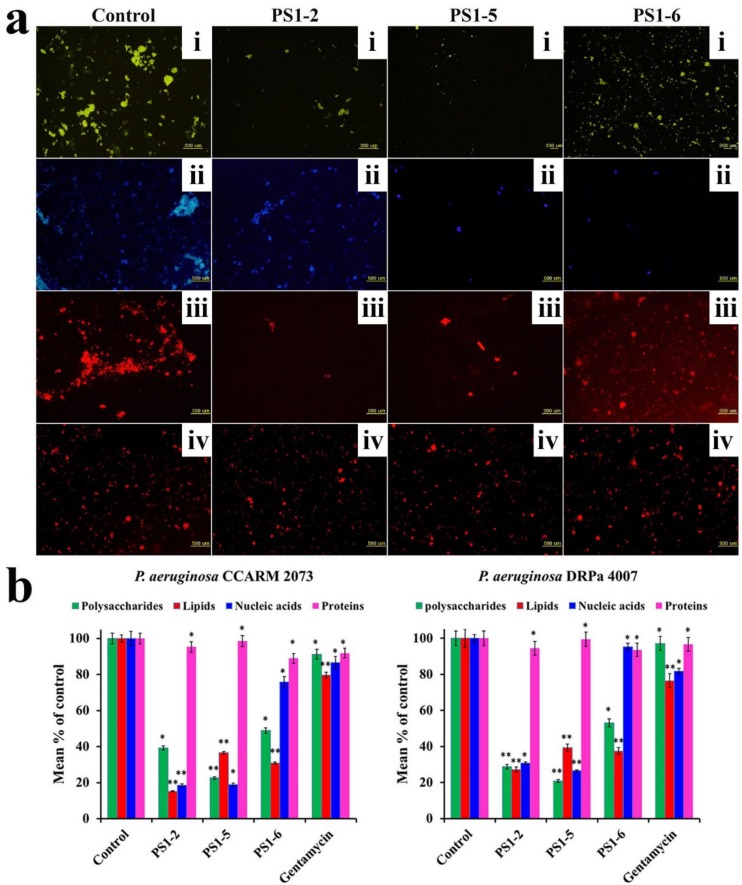
Eliminative action of PS peptides in biofilm extracellular polymeric substances (EPSs). (**a**) EPS preformed by *P. aeruginosa* DRPa 4007 was stained by fluorescein isothiocyanate-labeled concanavalin A (FITC-ConA) for carbohydrates (**i**), DAPI for nucleic acids (**ii**), Nile red for lipids (**iii**), and SYPRO red for proteins (**iv**) and recorded using fluorescence microscopy. (**b**) Eliminative percentages of PS peptides for EPS components preformed by *P. aeruginosa* CCARM 2073 and DRPa 4007 were evaluated by measuring fluorescence intensity. * Indicates statistical significance compared to the control (* *p* < 0.05; ** *p* < 0.001).

**Figure 5 molecules-24-04560-f005:**
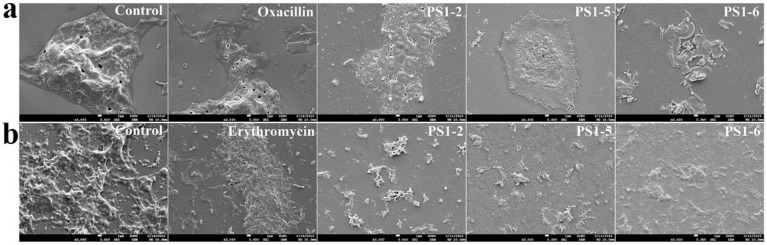
Reductive effects of PS peptides on biofilm preformed by *P. aeruginosa* DRPa 4007 (**a**) and *S. aureus* DRSa 3125 (**b**) on plastic disks. After biofilm formation, antibiotics or PS peptides were applied at their MICs for 24 h.

**Figure 6 molecules-24-04560-f006:**
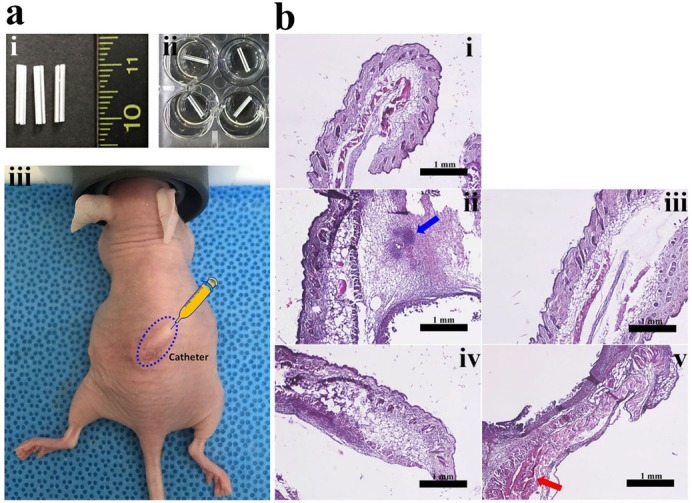
In vivo anti-biofilm study of PS1 peptides in the back skin of nude mice. (**a**) Intravenous 16G catheters (**i**) cut to 1 cm were pre-incubated in 24-well plates (**ii**) to which *S. aureus* DRSa 3125 was added for 36 h. Catheters with biofilm formation were implanted into mouse back skin (**iii**) and 1 mg/kg peptides were injected twice daily for two days. (**b**) Skin tissues were stained with hematoxylin and eosin and visualized using an inverted microscope. (**i**) control, (**ii**) catheter with DRSa 3125 biofilm, (**iii**) catheter with DRSa 3125 biofilm + PS1-2 peptide, (**iv**) catheter with DRSa 3125 biofilm + PS1-5 peptide, and (**v**) catheter with DRSa 3125 biofilm + PS1-6 peptide.

**Table 1 molecules-24-04560-t001:** Minimum inhibitory concentrations (MICs) of peptides and antibiotics against planktonic bacterial cells.

Bacteria	MIC (µM (µg/mL))					
	PS1-2	PS1-5	PS1-6	Gentamicin	Oxacillin	Erythromycin
***P. aeruginosa***						
ATCC 15692	2 (3.67)	2 (4)	2 (3.37)	1 (0.96)	4 (3.21)	-
CCARM 2073	2 (3.67)	2 (4)	2 (3.37)	256 (244)	512 (411)	-
CCARM 2075	2 (3.67)	2 (4)	1 (1.68)	256 (244)	512 (411)	-
DRPa 4007	4 (7.34)	2 (4)	2 (3.37)	512 (489)	256 (206)	-
DRPa 3241	2 (3.67)	2 (4)	2 (3.37)	512 (489)	128 (103)	-
***S. aureus***						
ATCC 25923	4 (7.34)	32 (64)	2 (3.37)	-	8 (6.42)	2 (1.47)
CCARM 3125	4 (7.34)	16 (32)	2 (3.37)	-	512 (411)	256 (188)
CCARM 3709	2 (3.67)	16 (32)	2 (3.37)	-	32 (25.7)	128 (93.9)
DRSa 3399	4 (7.34)	2 (4)	2 (3.37)	-	512 (411)	256 (188)
DRSa 3518	2 (3.67)	2 (4)	2 (3.37)	-	256 (206)	512 (376)

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
