# Peer review of "Anti-Biofilm Effects of Synthetic Antimicrobial Peptides Against Drug-Resistant Pseudomonas aeruginosa and Staphylococcus aureus Planktonic Cells and Biofilm"

_molecules, 2019, doi:10.3390/molecules24244560_

Round 1
Reviewer 1 Report
Park et coauthor describe the effect of synthetic peptides on Pseudomonas aeriginosa and Staphylococcus aureus biofilm. The subject is very interesting and the study is well thought out, however there are some points that need to be clarified
Major comments
The synthetic peptides studied have already been characterized in a previous work. In the introduction at least the rationale of the chosen sequences should be indicated and, if possible, a brief description of how they have been synthesized and purified. In the work it is not indicated which statistical analysis was used to verify the significance. The analysis should be described in materials and methods and in the caption of each figure the significance of the reported data, the number of experiments conducted and whether the measurements were performed in duplicate or tripled should be indicated. Only the time kill of Pseudomonas was reported in the work, why was not that of Staphylococcus reported? Provide at least one justification. In the description of the antibiofilm effect, the concentration of peptides and antibiotics tested for biofilm determination should be indicated. Furthermore, staining with the Crystal Violet offers a measurement of the biofilm mass but provides no indication of the phenotype (page 4, line 127). Animal experiments were conducted with too few mice. The number of animals studied must also be indicated in the statistical analysis. The increase in the thickness of the epidermis (page 8, line 180) should be reported. Peptides toxicity is an important factor in determining which peptides to test in vivo. In vitro cytotoxicity assay should be provided. For MIC determination an international protocol (CLSI) should be used.
Minor comments
In the title Staphylococcus Aureus should be replaced with Staphylococcus aureus, in the nomenclature the genre must be written in capital letters and the species must be written in lower case. The spelling of some words is not correct, revise the manuscript. The reference referring to the percentage of chronic biofilm-related infections is from 2007. a more recent report should be reported. Page 2, line 53: adhesion should be replaced with adhesion. Page 2, line 87: the reference 24 is not in numerical order. Throughout the manuscript the references to the figures have been indicated as (figures x (a)). Internal brackets can be eliminated for greater clarity (Figure xa). In EUCAST the threshold values are indicated in micrograms / ml. For better readability, the correspondence between uM and ug / ml should be reported in the text or in the caption of the figures for Gentamicin Oxacillin and Erythromycin. For a better understanding of the data, Figure 1 could be replaced with a table, or to avoid splitting the figure the MIC could be indicated in a logarithmic scale. Page 8, line 178: What is dual injection? Page 10, line237: the volume of the aliquots should be indicated. Page 10, lines 246 and 255: The authors should explain why a different concentration was used for biofilm culture. Page 11, line 268: The references reported [26-28] are not appropriated, probably the right references are [36-38]. For Reductive EPS analysis the concentration of reagent used anf the time of incubation should be indicated. In vivo study: the method for histology should be described.Author Response
Reviewer #1:
Park et coauthor describe the effect of synthetic peptides on Pseudomonas aeriginosa and Staphylococcus aureus biofilm. The subject is very interesting and the study is well thought out, however there are some points that need to be clarified
Major comments
Q1) The synthetic peptides studied have already been characterized in a previous work. In the introduction at least the rationale of the chosen sequences should be indicated and, if possible, a brief description of how they have been synthesized and purified.
A1) According to reviewer’ comment, followed description was added in the introduction: “In previous study, we reported a potent antimicrobial activity and membranolytic mechanism of a novel peptide series with repeated sequences of ‘XWZX’ (X: lysine or arginine, Z: leucine, tyrosine, valine, or glycine) [24]. Among eight peptides, PS1-2 [(KWYK)3] peptide with the highest therapeutic index (cell selectivity between bacteria and mammalian cells) was selected as an anti-biofilm target peptide in this study. To determine which of the lysine and arginine residues as a cationic amino acid is effective for anti-biofilm activity, PS1-5 [(RWYR)3] was compared. In addition, although PS1-5 [(KWLK)3] was found to be highly cytotoxic, it was chosen to compare the effects of aliphatic and aromatic side chains on anti-biofilm prevention. Three peptides synthesized by using microwave peptide synthesizer were purified by C18 reverse phase HPLC and were evaluated in vitro anti-biofilm activity on drug-susceptible and drug-resistant P. aeruginosa and S. aureus cells by in vitro phenotypic and EPS analyses, and in vivo evaluation using a catheter-implanted mouse model.”
Q2) In the work it is not indicated which statistical analysis was used to verify the significance. The analysis should be described in materials and methods and in the caption of each figure the significance of the reported data, the number of experiments conducted and whether the measurements were performed in duplicate or tripled should be indicated.
A2) We added and presented statistical analyses in method section and figure. Statistical analysis was calculated as mean values of at least three independent determinations ± SD and the P-values (Student t-test).
Q3) Only the time kill of Pseudomonas was reported in the work, why was not that of Staphylococcus reported? Provide at least one justification.
A3) We already performed time-killing kinetic assay in both P. aeruginosa and S. aureus strains, but we presented only P. aeruginosa strains because peptides exerted to same bacteriocidal kinetic against both bacterial strains. According to reviewer’s comment, we added killing-kinetic results in figure 2.
Q4) In the description of the antibiofilm effect, the concentration of peptides and antibiotics tested for biofilm determination should be indicated. Furthermore, staining with the Crystal Violet offers a measurement of the biofilm mass but provides no indication of the phenotype (page 4, line 127).
A4) We edited some descriptions. The term "phonotypic analyses" used in this paragraph means that the biofilm mass was measured by Crystal Violet staining. So I deleted this phrase.
Q5) Animal experiments were conducted with too few mice. The number of animals studied must also be indicated in the statistical analysis. The increase in the thickness of the epidermis (page 8, line 180) should be reported.
A5) The number of mice used in this study was 4 per of group (described in method section). Method for statistical analysis was added in method section.
Q6) Peptides toxicity is an important factor in determining which peptides to test in vivo. In vitro cytotoxicity assay should be provided.
A6) In vitro cytotoxicity assay for the peptides used in this study has been reported y hemolysis (erythrocytes) and cytotoxicity (HaCaT cells) assays in previous studies (reference 29). So, we do not add the data of in vitro cytotoxicity in this manuscript because of the use of data redundancy.
Q7) For MIC determination an international protocol (CLSI) should be used.
A7) As the reviewer comment, antimicrobial tests should be used international protocols such as NCCLS, but we use current methods because the use of acetic acid and BSA may provide artificial or additional effects in anti-biofilm assay. In addition, we have been used the current antimicrobial test methods for a long time and have reported in a number of research articles.
Minor comments
Q1) In the title Staphylococcus Aureus should be replaced with Staphylococcus aureus, in the nomenclature the genre must be written in capital letters and the species must be written in lower case.
A1) The submitted manuscript is correctly marked. This is due to a different version of the program (MS word).
Q2) The spelling of some words is not correct, revise the manuscript. The reference referring to the percentage of chronic biofilm-related infections is from 2007. a more recent report should be reported.
A2) We edited carefully and reference was changed to recent report.
Q3) Page 2, line 53: adhesion should be replaced with adhesion.
A3) We corrected it.
Q4) Page 2, line 87: the reference 24 is not in numerical order.
A4) We corrected it.
Q5) Throughout the manuscript the references to the figures have been indicated as (figures x (a)). Internal brackets can be eliminated for greater clarity (Figure xa).
A5) Internal brackets were removed.
Q6) In EUCAST the threshold values are indicated in micrograms / ml. For better readability, the correspondence between uM and ug / ml should be reported in the text or in the caption of the figures for Gentamicin Oxacillin and Erythromycin. µg/mL unit of MICs was added in table 1.
A6) The uM units in this paper are to compare the activities between peptides and antibiotics by not weight but the number of molecules.
Q7) For a better understanding of the data, Figure 1 could be replaced with a table, or to avoid splitting the figure the MIC could be indicated in a logarithmic scale.
A7) We replaced figure 1 to table 1.
Q8) Page 8, line 178: What is dual injection? Page 10, line237: the volume of the aliquots should be indicated.
A8) We corrected to “injection of peptides in twice daily for two days”. Volume was added to “50 µL”.
Q9) Page 10, lines 246 and 255: The authors should explain why a different concentration was used for biofilm culture.
A9) In this assay, the amount of the treated bacteria was increased by ten times, compared to the above inhibition assay for biofilm formation in order to form a biofilm completely. In addition, the preventing biofilm formation was to inhibit bacterial growth or biofilm formation factors, but it was required to remove the biofilm EPS and bacteria on the well surface. Therefore, a large amount of peptides was exposed in preformed biofilm.
Q10) Page 11, line 268: The references reported [26-28] are not appropriated, probably the right references are [36-38].
A10) We corrected it.
Q11) For Reductive EPS analysis the concentration of reagent used and the time of incubation should be indicated.
A11) We added concentrations of each reagent in method.
Q12) In vivo study: the method for histology should be described.
A12) We added the detailed method for histological analysis.
Reviewer 2 Report
The study is interesting and impactful. A few issues may be addressed to improved the manuscript:
1- The introduction section misses an important paper recently published on the anti-biofilm action of peptides (Journal of Antimicrobial Chemotherapy, Volume 74, Issue 9, 2617–2625, https://doi.org/10.1093/jac/dkz223)
2- Line 109: What line of evidence support eventual translocation of antimicrobial peptides via efflux pumps?
3- How can peptides remove lipids or saccharides from the extracellular matrix of biofilms? How credible/plausible is the hypothesis that peptides directly remove biofilm components?
4- How were SEM results quantified so conclusions in section 2.6 can be supported on evidence?
Author Response
Reviewer #2:
The study is interesting and impactful. A few issues may be addressed to improved the manuscript:
Q1) The introduction section misses an important paper recently published on the anti-biofilm action of peptides (Journal of Antimicrobial Chemotherapy, Volume 74, Issue 9, 2617–2625, https://doi.org/10.1093/jac/dkz223)
A1) We added this reference.
Q2) Line 109: What line of evidence support eventual translocation of antimicrobial peptides via efflux pumps?
A2) This is our mistake. We deleted it.
Q3) How can peptides remove lipids or saccharides from the extracellular matrix of biofilms? How credible/plausible is the hypothesis that peptides directly remove biofilm components?
A3) Peptides possess amphipathic and alpha-helical structures, and cationicity. So, cationic peptide can bind to anionic lipid head groups, polysaccharides, and nucleic acids via electrostatic interaction. Moreover, their hydrophobic parts can bind strongly to lipid tail groups and carbohydrates. Peptide can be diffused into biofilm and network structures of EPS can break down by these affinities for each component.
Q4) How were SEM results quantified so conclusions in section 2.6 can be supported on evidence?
A4) Although the thickness of the biofilm can be analyzed quantitatively by using SEM, it is very difficult to adjust because the measuring plane must be parallel to the observation point. Therefore, most of the SEM results can be analyzed comparably.
Round 2
Reviewer 1 Report
The authors have modified and corrected all the parts found critical in the first revision. The manuscript is much improved.